# The Interaction of the Senescent and Adjacent Breast Cancer Cells Promotes the Metastasis of Heterogeneous Breast Cancer Cells through Notch Signaling

**DOI:** 10.3390/ijms22020849

**Published:** 2021-01-15

**Authors:** Na Zhang, Jiafei Ji, Dandan Zhou, Xuan Liu, Xinglin Zhang, Yingqi Liu, Weifang Xiang, Meida Wang, Lian Zhang, Guannan Wang, Baiqu Huang, Jun Lu, Yu Zhang

**Affiliations:** 1The Key Laboratory of Molecular Epigenetics of Ministry of Education (MOE), Northeast Normal University, Changchun 130024, China; zhangn@nenu.edu.cn (N.Z.); zhoudd150@nenu.edu.cn (D.Z.); zhangxl791@nenu.edu.cn (X.Z.); wangmd461@nenu.edu.cn (M.W.); wanggn258@nenu.edu.cn (G.W.); 2The Institute of Genetics and Cytology, Northeast Normal University, Changchun 130024, China; Jijiafei1109@gmail.com (J.J.); liux468@nenu.edu.cn (X.L.); liuyq273@nenu.edu.cn (Y.L.); xiangwf982@nenu.edu.cn (W.X.); zhanglian1125@163.com (L.Z.); huangbq@nenu.edu.cn (B.H.); luj809@nenu.edu.cn (J.L.)

**Keywords:** doxorubicin, co-culture system, breast cancer cells senescence, Notch signaling, EMT

## Abstract

Chemotherapy is one of the most common strategies for tumor treatment but often associated with post-therapy tumor recurrence. While chemotherapeutic drugs are known to induce tumor cell senescence, the roles and mechanisms of senescence in tumor recurrence remain unclear. In this study, we used doxorubicin to induce senescence in breast cancer cells, followed by culture of breast cancer cells with conditional media of senescent breast cancer cells (indirect co-culture) or directly with senescent breast cancer cells (direct co-culture). We showed that breast cancer cells underwent the epithelial–mesenchymal transition (EMT) to a greater extent and had stronger migration and invasion ability in the direct co-culture compared with that in the indirect co-culture model. Moreover, in the direct co-culture model, non-senescent breast cancer cells facilitated senescent breast cancer cells to escape and re-enter into the cell cycle. Meanwhile, senescent breast cancer cells regained tumor cell characteristics and underwent EMT after direct co-culture. We found that the Notch signaling was activated in both senescent and non-senescent breast cancer cells in the direct co-culture group. Notably, the EMT process of senescent and adjacent breast cancer cells was blocked upon inhibition of Notch signaling with *N*-[(3,5-difluorophenyl)acetyl]-l-alanyl-2-phenyl]glycine-1,1-dimethylethyl ester (DAPT) in the direct co-cultures. In addition, DAPT inhibited the lung metastasis of the co-cultured breast cancer cells in vivo. Collectively, data arising from this study suggest that both senescent and adjacent non-senescent breast cancer cells developed EMT through activating Notch signaling under conditions of intratumoral heterogeneity caused by chemotherapy, which infer the possibility that Notch inhibitors used in combination with chemotherapeutic agents may become an effective treatment strategy.

## 1. Introduction

Chemotherapy is a type of clinical cancer treatment involving the systemic administration of genotoxic compounds that induce cancer cell death via establishing DNA damage response signaling networks. However, in many cases, tumor recurrence occurs at the original or distant sites [1,2]. Chemotherapy can trigger increased intratumoral heterogeneity [3] due to limitations in tissue permeability and variable concentrations of drug exposure of tumor cells, and this facilitates the formation of a gradient decrease with increasing distance between tumor cells and blood vessels after drugs enter solid tumors through blood [4,5]. Accordingly, chemotherapeutic drugs induce different fates of tumor cells at distinct solid tumor locations. Tumor cells close to blood vessels undergo apoptosis due to exposure to lethal concentrations of the drugs, while those at a moderate distance from blood vessels experience senescence. Tumor cells located far away enough from the vessels are not sensitive to chemotherapy drugs and potentially become seeds of tumor recurrence [6,7,8]. Ultimately, the roles of senescent tumor cells induced by chemotherapy drugs in tumor progression remain unclear at present.

Cell senescence, an irreversible state of cell proliferation, is an important cytological mechanism for body development and for the prevention of cancer [9,10]. Previous studies have demonstrated that senescent cells display a senescence-associated secretory phenotype (SASP) defined by an aptitude to secrete a variety of cytokines, growth factors, and enzymes that are expected to alter the surrounding microenvironment [11,12]. SASP is a double-edged sword, which can reinforce the senescent phenotype in both an autocrine and paracrine manners, and activate the immune clearance of senescent cells from tissues, thereby contributing to tumor suppression [13,14,15]. SASP has also been reported to play a tumorigenic role, promoting cellular growth and the epithelial–mesenchymal transition (EMT) [16,17]. Other than SASP, the effects and mechanisms of action of senescent cells on tumor cells are yet to be established.

The Notch signaling pathway, which is evolutionarily conserved and involved in a range of developmental and physiological processes, facilitates direct cell–cell communication [18,19]. This pathway displays either oncogenic or tumor-suppressive functionality depending on the type of tissues and their context [20]. The pathway is activated by interactions between Notch receptors and ligands (Jagged, JAG and Delta-like family of ligands) on the adjacent cell surface. Notch receptors undergo a series of proteolytic cleavage events liberating the intracellular domain (ICD) from the nucleus, which, together with transcriptional co-activators, drives the transcription of target genes, including the HES and HEY transcription factor families [18]. The Notch pathway is also actively involved in regulation of EMT [21,22]. Recent studies suggest that Notch signaling is closely related to the process of cell senescence, and decrease in Notch activation promotes senescence [23,24,25]. In addition, Notch signaling is associated with tumor resistance to chemotherapy [26,27,28,29], but further studies are required to elucidate the underlying mechanisms of this action.

In the present study, to explore the effect of tumor senescence induced by chemotherapeutic drugs on tumor progression, we established a direct co-culture model of breast cancer cells with drug-mediated senescent breast cancer cells. Our data highlight a novel mechanism underlying tumor recurrence following chemotherapy whereby the drug-induced breast cancer cell senescence promotes the EMT of adjacent tumor cells through activating Notch signaling. These results are in support of the application of Notch inhibitors as a promising adjuvant therapy to doxorubicin for breast cancer.

## 2. Results

### 2.1. Senescence of Breast Cancer Cells Induced by Doxorubicin Promotes EMT of Adjacent Breast Cancer Cells

To explore the effects of doxorubicin-induced senescent breast cancer cells on the tumorigenesis process, we established direct and indirect co-culture systems. We initially induced the senescence of MCF-7 breast cancer cells (SEN-MCF-7) by treatment with doxorubicin (200 nM; Appendix A). In the direct co-culture system, MCF-7-mRFP (fused to red fluorescent protein, mRFP) and SEN-MCF-7 cells were co-cultured in Dulbecco’ Modified Eagle Media (DMEM) medium (Figure 1A, left, and Appendix A). In the indirect co-culture system, MCF-7-mRFP cells were cultured in media that contains the senescent culture medium of the SEN-MCF-7 (SEN-CM-MCF-7-mRFP) (Figure 1A, right). Previous studies have shown that SASP contributes to tumor progression by promoting EMT in neighboring immortalized or transformed epithelial cells [16,17]. Here, we initially analyzed the effects of both culture methods on the EMT process of MCF-7 breast cancer cells. We found that SEN-CM-MCF-7-mRFP cells underwent EMT, accompanied by downregulation of the epithelial marker, E-cadherin, and upregulation of the mesenchymal marker, Vimentin. Compared with SEN-CM-MCF-7-mRFP, adjacent breast cancer cells co-cultured with SEN-MCF-7 cells (Co-MCF-7-mRFP) exhibited more significant downregulation of E-cadherin and upregulation of Vimentin (Figure 1B,C). Consistently, breast cancer MCF-7-mRFP cells gained the ability to migrate and invade both indirect and indirect co-culture systems. Notably, Co-MCF-7-mRFP cells showed stronger migration and invasion abilities (Figure 1D–G). Our results indicated that senescent breast cancer cells can promote the EMT of adjacent breast cancer cells.

### 2.2. Senescent Breast Cancer Cells Re-Enter the Cell Cycle and Undergo EMT after Direct Co-Culture with Breast Cancer Cells

Our initial experiments (Figure 1) showed that the senescent breast cancer cells promoted the EMT of adjacent breast cancer cells in direct co-culture systems. To explore the effects of adjacent breast cancer cells (MCF-7-mRFP) on senescent tumor cells, senescent breast cancer cells (Co-SEN-MCF-7) were obtained after direct co-culture with breast cancer cells (MCF-7-mRFP) (Figure 2A), and the activity of senescence-associated β-galactosidase (SA-β-gal) in Co-SEN-MCF-7 and SEN-MCF-7 cells was examined. The SA-β-gal activity of Co-SEN-MCF-7cells was significantly decreased compared with that in SEN-MCF-7cells (Figure 2B,C), which was accompanied by cyclinA2 upregulation after direct co-culture for 3 or 6 days (Figure 2D). The phenomenon prompted us to investigate whether the cell cycle of senescent breast cancer cells was altered after co-culture with breast cancer cells. Flow cytometry analysis of the DNA content revealed the accumulation of the G2/M phase SEN-MCF-7 cells, in comparison with MCF-7 cells. Senescent breast cancer cells (Co-SEN-MCF-7) in the G2/M phase were released from cycle arrest state after direct co-culture for 6 days compared with SEN-MCF-7 cells (Figure 2E), signifying the re-entry of Co-SEN-MCF-7 into the cell cycle. Moreover, we have found that the senescent breast cancer cells re-enter into the cell cycle after direct co-culture with non-senescent breast cancer cells. Unexpectedly, Co-SEN-MCF-7 cells also regained migration and invasion ability (Figure 2F,G), which is indicative of EMT progression. Indeed, the adhesion of Co-SEN-MCF-7 cells became weaker after direct co-culture, which was accompanied by decreased E-cadherin and increased Vimentin expression, compared with SEN-MCF-7 cells (Figure 2H,I). Thus, our results suggest that senescent breast cancer cells re-enter the cell cycle and undergo EMT after direct co-culture with breast cancer cells.

### 2.3. Non-Cell Autonomous Activation of the Notch Signal in Senescent and Adjacent Breast Cancer Cells

We have confirmed that direct co-culture promoted breast cancer invasion and migration to a greater extent than indirect co-culture did (Figure 1C–F). Since Notch signaling depends on direct contact between two adjacent cells, we examined the changes in Notch signaling in direct co-cultures (Figure 3A). Our results revealed a significant upregulation of cleaved, active Notch1 intracellular domain (N1ICD) and canonical Notch1target HES1 in adjacent breast cancer cells (Co-MCF-7-mRFP) of SEN-MCF-7, compared with that in MCF-7-mRFP cultured with senescence culture medium (SEN-CM-MCF-7-mRFP) (Figure 3B). Similarly, the Notch target genes, *HES1* and *HEY1*, as well as *JAG1* (which encodes a ligand of Notch), were significantly upregulated in Co-MCF-7-mRFP relative to that in SEN-CM-MCF-7-mRFP cells (Figure 3C). Meanwhile, Notch signaling was reactivated in senescent breast cancer cells (Co-SEN-MCF-7), which was accompanied by the downregulation of N1ICD and HES1, after direct co-culture with breast cancer cells (Figure 3D). Consistently, *HES1*, *HEY1*, and *JAG1* were significantly upregulated in Co-SEN-MCF-7 cells in the direct co-culture setting (Figure 3E). Moreover, Notch signaling was reactivated in direct co-cultured T47D cells (Figure 4G). Clearly, these findings suggest the activation of Notch signaling in both senescent and non-senescent breast cancer cells under direct co-culture conditions.

### 2.4. Inhibition of Notch Signaling Impairs EMT Progression and Inhibits Metastasis of the Senescent and Adjacent Non-Senescent Breast Cancer Cells

To further confirm the involvement of Notch signaling in the non-autonomous regulation of EMT between breast cancer and senescent cells, breast cancer cells were treated with the γ-secretase inhibitor of Notch, *N*-[(3,5-difluorophenyl)acetyl]-l-alanyl-2-phenyl]glycine-1,1-dimethylethyl ester (DAPT), under the direct co-culture system (Figure 4A and Appendix A). Notch signaling of adjacent breast cancer cells (Co-MCF-7-mRFP) and senescent breast cancer cells (Co-SEN-MCF-7) treated with DAPT was evidently suppressed in the co-culture system, which was accompanied by N1ICD downregulation (Figure 4B). Meanwhile, the epithelial marker E-cadherin was significantly upregulated, while the mesenchymal marker Vimentin was dramatically downregulated in Co-MCF-7-mRFP and Co-SEN-MCF-7 (Figure 4B). In addition, Co-MCF-7-mRFP and Co-SEN-MCF-7 cells exhibited a marked decrease in migratory and invasive behavior after treatment with DAPT (Figure 4C–F). Moreover, we got similar results in another breast cancer T47D cells (Figure 4G–K and Appendix A). To further evaluate the critical role of the interaction through activating Notch signaling between senescent and non-senescent breast cancer cells in breast cancer metastasis, we built the direct co-culture system with MCF-7-luciferase (ectopically expressing the luciferase, MCF-7-luc) cells and got similar results as above (Appendix A). Consistently, we found that DAPT inhibited the metastasis of the co-cultured breast cancer cells in vivo (Figure 4L,M). These data are strongly indicative of the pivotal role of Notch signaling between senescent and non-senescent breast cancer cells in promoting breast cancer metastasis.

## 3. Discussion

In this study, we unveiled a regulatory mechanism underlying the chemotherapeutic drug-mediated breast cancer recurrence that involved the activation of Notch signaling. Our data showed that senescent breast cancer cells induced by doxorubicin treatment promoted Notch signaling in adjacent breast cancer cells. Conversely, breast cancer cells stimulated Notch signaling in adjacent senescent breast cancer cells. Upon activation of Notch signaling, both senescent and non-senescent breast cancer cells underwent EMT, which accelerated migration and invasion. More importantly, we discovered that the enhancement of breast cancer cell migration and invasion ability by chemotherapeutic drug-induced breast cancer cell senescence can be inhibited by the Notch inhibitor DAPT. Meanwhile, the inhibition of Notch signaling inhibited breast cancer metastasis in vivo (Figure 5).

The non-autonomous functionality of senescent cells is thought to be attributed predominantly to the SASP that can alter the surrounding microenvironment in a paracrine manner [11,12], and induce the EMT of tumor cells [10,17,30]. Previous studies on 4-hydroxytamoxifen (4-OHT)-mediated senescent fibroblasts have shown that Notch1 is activated in the early stages of cell senescence and suppressed when cells are fully senescent [25]. To our knowledge, the present study is the first attempt to prove that senescent breast cancer cells promote EMT through activating Notch signaling, which can be transmitted between senescent and adjacent breast cancer cells (Figure 3 and Figure 4). Notably, breast cancer cells showed stronger migration and invasion capacity in the direct co-culture system compared with that in senescent conditional cultures (Figure 1). Presumably, this may be explained by the simultaneous activity of senescent breast cancer cells in stimulating the Notch signaling pathway of neighboring tumor cells and secreting SASP.

Cell senescence is a relatively stable state of cell cycle arrest and growth stagnation that maintains low metabolic activity but continued survival [31,32]. In this study, we showed that the SA-β-gal activity was decreased significantly, with a concurrent upregulation of cyclinA2 in senescent breast cancer cells co-cultured with breast cancer cells (Figure 2B–D and Figure 4G). Additionally, we have provided evidence that senescence breast cancer cells re-entered the cell cycle only after direct co-culture with breast cancer cells (Figure 2E). Previous studies have demonstrated that senescent cancer cells induced by chemotherapy can restore the self-renewal capacity of tumor cells and possess the stem-related metastatic and tumorigenic abilities in vivo [33]. Notch and canonical Wntsignalings activated in therapy-induced senescence are essential drivers of the enhanced tumor initiation capacity of senescence-released tumor cells [33]. However, senescent breast cancer cells returned to the cell cycle independent of Notch signaling after direct co-culture with active breast cancer cells (Appendix A). The Wnt signaling of senescent cells was repressed and re-activated after co-culture with non-senescent cells accompanied by upregulation of the canonical Wnt targets, β-catenin, cyclinD1, and c-MYC (Appendix A). Accordingly, we speculate that re-entry of senescent breast cancer cells into the cell cycle was associated with the Wnt signaling pathway.

Doxorubicin is an anthracycline antibiotic with strong anticancer activity that prevents DNA replication and RNA synthesis, resulting in cellular apoptosis [34]. Doxorubicin is a commonly used drug for cancer treatment. Recent studies have shown that doxorubicin causes the senescence of synovial sarcoma cells without apoptosis, and this process is accompanied by the abnormal expression of senescence-related marker genes and increased β-gal activity [35]. Doxorubicin is also reported to activate the p53/p21 signaling pathway, block cell cycle progression, and cause senescence in breast cancer cells [36]. In addition, available data strongly suggest that chemotherapy triggers the intratumoral heterogeneity [3]. Therefore, tumor recurrence may probably be associated with tumor heterogeneity mediated by chemotherapeutic drugs. As shown in our experiments, both senescent and adjacent active MCF-7 breast cancer cells acquired mesenchymal characteristics through the activation of Notch signaling (Figure 1 and Figure 3). It has been reported that Notch1 inhibitors alter the cancer stem cell (CSC) phenotype and reduce the formation of brain metastasis from breast cancer [37]. In our study, the inhibition of Notch signaling suppressed the metastasis of senescent and adjacent non-senescent breast cancer cells in vivo (Figure 4). Collectively, based on these findings, we propose that a combination of use of Notch inhibitors and chemotherapeutic drugs may offer a more effective therapeutic strategy.

In conclusion, we have identified the activation of Notch signaling as a mechanism underlying breast cancer recurrence mediated by chemotherapy drugs. Our experimental evidence contributes to further understanding of breast cancer recurrence following chemotherapy and supports the utility of inhibitors of the Notch pathway as promising adjuvant agents in combination with doxorubicin for breast cancer.

## 4. Materials and Methods

### 4.1. Cell Cultures

Cell lines (MCF-7, T47D and Phoenix) were obtained from the American Type Culture Collection (ATCC, Manassas, VA, USA)and characterized by DNA fingerprinting and isozyme detection. Cells were tested by a MycoBlue Mycoplasma Detector (Vazyme Biotech, Nanjing, China) to exclude Mycoplasma contamination before experiments. MCF-7, T47D, and Phoenix cells were cultured in DMEM (Sigma-Aldrich, St. Louis, MO, USA) containing 10% Fetal Bovine Serum (FBS). All cells were grown at 37 °C with 5% CO_2_ and passaged.

#### 4.1.1. Direct Co-Culture

MCF-7 (3,000,000) cells were seeded in a 10 cm dish and incubated for 24 h with 200 nM Doxorubicin in DMEM with 10% FBS. Subsequently, cells were continually incubated for 3 days in fresh medium. Cultures were setup at a cell ratio of 3:1 (senescent breast cancer MCF-7 cells: non-senescent breast cancer MCF-7 cells) and seeded at a density of 3,000,000 cells/well in a 10 cm dish for 3 or 6 days before analysis.

#### 4.1.2. Indirect Co-Culture

Medium of senescent MCF-7 cells, designated senescent conditioned medium (SEN-CM), was collected, centrifuged at 5000× *g*, and filtered through a 0.2 μm pore filter. SEN-CM was mixed with fresh medium at a 3:1 ratio for MCF-7-mRFP cell culture.

### 4.2. Reagents and Antibodies

Doxorubicin (Dox) and *N*-[(3,5-Difluorophenyl)acetyl]-l-alanyl-2-phenyl]glycine-1,1-dimethylethyl ester (DAPT) were obtained from Selleck (Houston, TX, USA). Antibodies were procured against E-cadherin (BD Biosciences, Lexington, KY, USA), Vimentin (BD Biosciences), β-actin (Sigma-Aldrich, A5228), Notch1 (Cell Signaling Technology, Boston, MA, USA), cleaved Notch1 (Cell Signaling Technology), cyclinA2 (Abcam, Cambridge, UK), Ki67 (GeneTex, GTX16667), Hes1 (Cell Signaling Technology), p21 (Proteintech, Rosemont, IL, USA), β-catenin (BD Biosciences), cyclinD1 (Santa Cruz Biotechnology, CA, USA), and c-Myc(Santa Cruz Biotechnology). Secondary goat anti-mouse and goat anti-rabbit antibodies were obtained from ZSGB-BIO (Beijing, China).

### 4.3. Plasmid and Retroviral Infection

pLPC-mRFP-puro expression vector and packaging vector Env.A were used in this study. The retrovirus packaging vector used was Env.A The generation of retrovirus in Phoenix cells and transfection of retroviral constructs into recipient cell lines were performed according to the manufacturer’s instructions (Invitrogen, Carlsbad, CA, USA).

### 4.4. Reverse Transcription, PCR and Real-Time PCR

Reverse transcription, PCR and real-time PCR were performed as described [38]. The primer sequences for PCR were as follows (5′–3′, sense, antisense): *β-actin* forward: 5′-GAGCACAGAGCCTCGCCTTT-3′ and reverse: 5′-ATCCTTCTGACCCATGCCCA-3′, HES1 forward: 5′-ACGTGCGAGGGCGTTAATAC-3′ and reverse:5′-ATTGATCTGGGTCATGCAGTTG-3′, *HEY1* forward: 5′-CCGCTGATAGGTTAGGTCTCATTTG-3′ and reverse: 5′-TCTTTGTGTTGCTGGGGCTG-3′, and *JAG1* forward: 5′-TGGTCAACGGCGAGTCCTTTAC-3′ and reverse: 5′-GCAGTCATTGGTATTCTGAGCACAG-3′.

### 4.5. Western Blot

Cells were lysed in 1 × Laemmlisample buffer after washing twice in cold Phosphate Buffered Saline without calcium and magnesium (PBS). Protein concentrations were determined using the bicinchoninic acid (BCA) protein assay. Protein lysates were subjected to SDS-PAGE, transferred to 0.45μm poresize Hydrophobic PVDF Transfer Membrane (Merck Millipore, Cork, Irland), detected with the appropriate primary antibodies coupled with HRP-conjugated corresponding secondary antibodies, and visualized using Enhanced chemiluminescenceECL reagent (GE Healthcare, Buckinghamshire, UK). The Tanon 5500 high-definition low-illumination CCD system was used to capture chemiluminescent signals.

### 4.6. Immunofluorescence

Immunofluorescence was performed as described [39].

### 4.7. Wound Healing and Transwell Migration

Wound healing and Transwell migration was performed as described [38].

### 4.8. SA-β-Gal Staining

SA-β-gal staining was performed as described [39].

### 4.9. Flow Cytometry

For the cell cycle assay, ethanol-fixed MCF-7 cells were analyzed for cellular DNA content after propidium iodide staining. Flow cytometric analysis was carried out using a fluorescence-activated cell sorting (FACS) Aria-II instrument. Data were obtained from 1,000,000 to 3,000,000 events for each sample. Analysis was performed using FACS Diva software. Signal amplification was decreased to normalize the dot plot for analysis and the amount of compensation was used to exclude overlap between the two signals.

### 4.10. In Vivo Mouse Lung Metastasis Assay

Detailed descriptions were presented before [38]. Briefly, MCF-7-luc, SEN-MCF-7-luc, and Co-culture-MCF-7-luc cells (500,000 cells suspended with 100 μL of PBS) treatment with or without DAPT were injected into the tail veins of female BALB/c nude mice (HFK Bioscience, Beijing, China) at 5 weeks old. Four weeks later, the lung metastases of mice were assessed with bioluminescence imaging, and lungs were fixed in Bouin’s solution. Then embedded lungs in paraffin and performed with H&E staining. All animal experiments were approved by the Animal Care Committee of the Northeast Normal University, China (Reference assurance number AP2013011).

### 4.11. Statistical Analysis

Results were compiled from at last three independent replicate experiments and presented as mean ± SD. The paired Student’s *t*-test (two-tailed) was used to calculate the significance of differences between groups. Data were considered significant at *p* < 0.05. Statistical analysis was conducted using GraphPad Prism software (GraphPad Software, La Jolla, CA, USA).

## Figures and Tables

**Figure 1 ijms-22-00849-f001:**
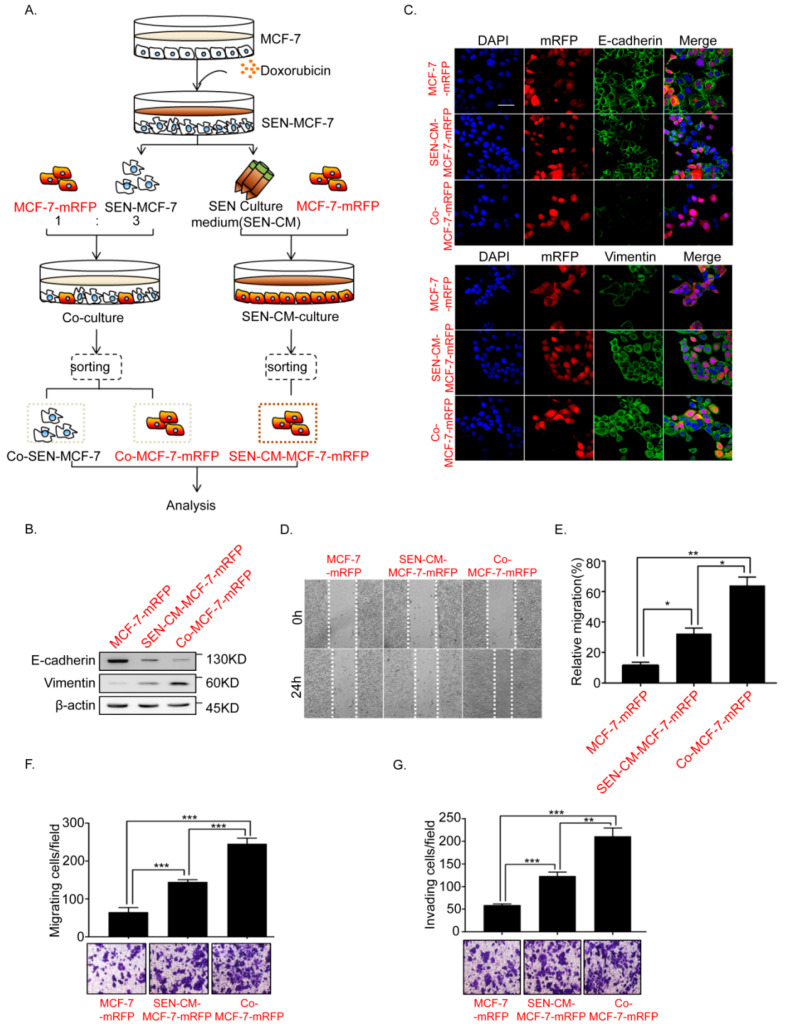
Doxorubicin (Dox)-induced MCF-7 cell senescence promoted the epithelial–mesenchymal transition (EMT) of adjacent breast cancer cells. (**A**) Schematic diagram showing the experimental setup. Schematic diagram of the co-culture systems. Senescent MCF-7 cells (SEN-MCF-7) induced by Dox (200nM) were co-cultured with MCF-7-mRFP at a 3:1 ratio (SEN-MCF-7:MCF7-mRFP). Senescent MCF-7 cells (SEN-MCF-7) were recovered from the supernatant and mixed with normal medium at a 3:1 ratio (senescence supernatant: normal medium) as conditional medium (SEN-CM) for MCF7-mRFP cell monocultures. Red and black letters, used to distinguish two types of MCF-7 cells. Red letters: MCF-7 cells fused to red fluorescent protein, mRFP. Black letters: MCF-7 cells without fusing to mRFP. (**B**) Immunoblot analysis of protein expression of the epithelial marker E-cadherin and mesenchymal marker Vimentin in MCF-7-mRFP cells with the indicated treatments. (**C**) Immunofluorescence staining for EMT markers in MCF-7-mRFP cells with the indicated treatments. Scale bar, 50μm. (**D**,**E**) Wound-healing assay of MCF-7-mRFP cells with the indicated treatments. (**D**) Representative micrographs of wound-healing assay are shown. (**E**) Data represent the relative migration ratio of cells per field (error bars indicate mean SD, *n* = 3 experimental replicates, * *p* < 0.05, ** *p* < 0.01, *** *p* < 0.001). (**F**) Migration assay of MCF-7-mRFP cells with the indicated treatments. Representative micrographs of the migrated cells are shown. Data represent the number of cells derived from mean cell counts of five fields (Error bars indicate mean SD, *n* = 3 experimental replicates, * *p* < 0.05, ** *p* < 0.01, *** *p* < 0.001). (**G**) Invasion assay of MCF-7-mRFP cells with the indicated treatments. Representative micrographs of the migrated cells are shown. Data represent the number of cells derived from mean cell counts of five fields (Error bars indicate mean SD, *n* = 3 experimental replicates, * *p* < 0.05, ** *p* < 0.01, *** *p* < 0.001).

**Figure 2 ijms-22-00849-f002:**
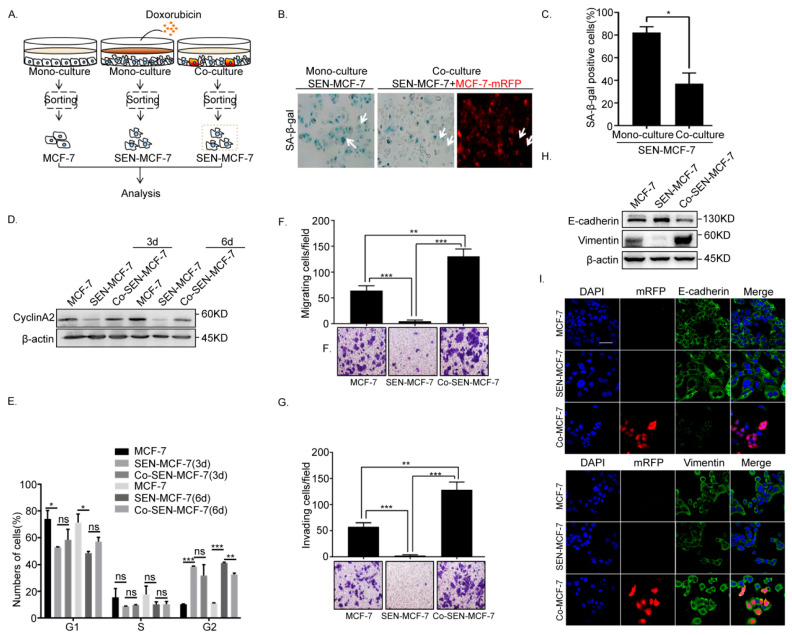
Senescent MCF-7 cells re-entered the cell cycle and underwent EMT in the co-culture. (**A**) Left: Scheme showing the experimental setup. Flow cytometry sorting of collected senescence breast cancer cells (Co-SEN-MCF-7) co-cultured with MCF-7-mRFP for 3 or 6 days. (**B**,**C**) β-Gal staining analysis of MCF-7 cells subjected to the indicated treatments and fluorescence microscopy of MCF-7-mRFP cells. Arrow, senescent MCF-7 cells (SEN-MCF-7 and Co-SEN-MCF-7). (**B**) Representative images. (**C**) Percentage of β-gal positive cells (error bars indicate the mean SD, *n* = 3 experimental replicates, * *p* < 0.05). (**D**) Right: Immunoblot analysis of protein expression of cyclinA2 in MCF-7 cells with the indicated treatments. (**E**) Cell cycle analysis of MCF-7 cells with the indicated treatments. Percentages of cell subpopulations at different cell cycle phases based on triplicate experiments (Error bars indicate mean SD, *n* = 3 experimental replicates, * *p* < 0.05, ** *p* < 0.01, *** *p* < 0.001). (**F**) Migration assays of MCF-7 cells with the indicated treatments. Representative micrographs of migrated cells are shown. Data represent the number of cells derived from mean cell counts of five fields (error bars indicate mean SD, *n* = 3 experimental replicates, ** *p* < 0.01, *** *p* < 0.001). (**G**) Invasion assay of MCF-7 cells with the indicated treatments. Representative micrographs of migrated cells are shown. Data represent the number of cells derived from mean cell counts of five fields (Error bars indicate mean SD, *n* = 3 experimental replicates, ** *p* < 0.01, *** *p* < 0.001). (**H**) Immunoblot analysis of protein expression of the epithelial marker, E-cadherin, and mesenchymal marker, Vimentin, in MCF-7 cells subjected to the indicated treatments. (**I**) Immunofluorescence staining of EMT markers in MCF-7 cells subjected to the indicated treatments. Scale bar, 50 μm.

**Figure 3 ijms-22-00849-f003:**
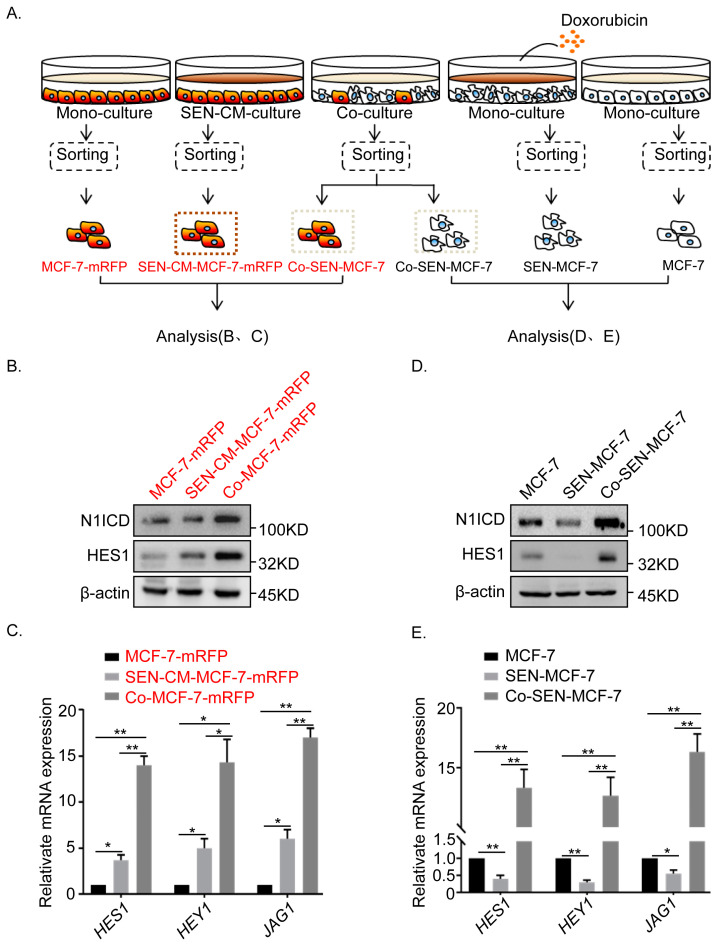
Activation of Notch signaling in both senescent and adjacent non-senescent breast cancer MCF-7 cells. (**A**) Scheme showing the experimental setup. Flow cytometry sorting of collected breast cancer cells after co-culture or monoculture. (**B**) Immunoblot analysis of protein expression of active Notch1 intracellular domain (N1ICD) and the canonical Notch target, HES1, assessed in MCF7-mRFP cells subjected to the indicated treatments. (**C**) RT-qPCR analysis of expression of target genes *HES1* and *HEY1* and *JAG1* in MCF7-mRFP cells subjected to the indicated treatments (error bars indicate mean SD, *n* = 3 experimental replicates,* *p* < 0.05, ** *p* < 0.01). (**D**) Immunoblot analysis of protein expression of active Notch1 intracellular domain (N1ICD) and the canonical Notch1target, HES1, assessed in MCF-7 cells subjected to the indicated treatments. (**E**) RT-qPCR analysis of gene expression of target genes HES1 and HEY1 and the JAG1 ligand of Notch in MCF-7 cells subjected to the indicated treatments (error bars indicate the mean SD, *n* = 3 experimental replicates,* *p* < 0.05, ** *p* < 0.01).

**Figure 4 ijms-22-00849-f004:**
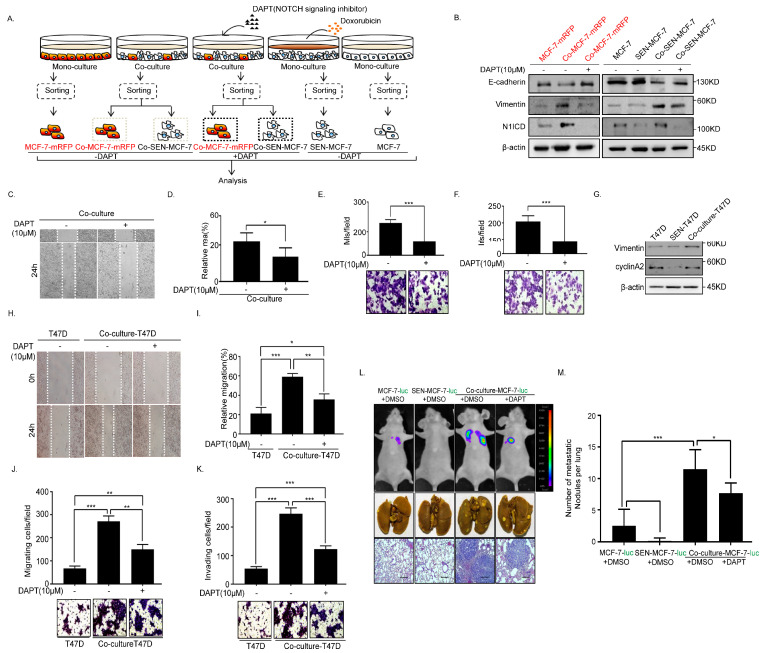
*N*-[(3,5-Difluorophenyl)acetyl]-l-alanyl-2-phenyl]glycine-1,1-dimethylethyl ester (DAPT) inhibits EMT and metastasis of the senescent and adjacent non-senescent breast cancer cells. (**A**) Scheme showing the experimental setup. Flow cytometry analysis of collected breast cancer cells in co-cultures or monocultures treated with or without DAPT (10 μM). (**B**) Immunoblot analysis of protein expression of active Notch1 intracellular domain (N1ICD), E-cadherin, and Vimentin in MCF-7-mRFP and MCF-7 cells treated with or without DAPT. (**C**,**D**) Wound-healing assay of MCF-7 cells (MCF-7-mRFP and SEN-MCF-7) treated with or without DAPT. (**C**) Representative micrographs of the wound-healing assay are shown. (**D**) Data represent the relative migration ratio of cells per field (error bars indicate mean SD, *n* = 3 experimental replicates, * *p* < 0.05). (**E**) Migration assay of MCF-7 cells (MCF-7-mRFP and SEN-MCF-7) treated with or without DAPT. Representative micrographs of migrated cells are shown. Data represent the number of cells derived from mean cell counts of five fields (error bars indicate mean SD, *n* = 3 experimental replicates, *** *p* < 0.001). (**F**) Invasion assay of MCF-7 cells (MCF-7-mRFP and SEN-MCF-7) with indicated treatments. Representative micrographs of the migrated cells are shown. Data represent the number of cells derived from mean cell counts of five fields (error bars indicate mean SD, *n* = 3 experimental replicates, * *p* < 0.05, ** *p* < 0.01, *** *p* < 0.001). (**G**) Immunoblot analysis of protein expression of E-cadherin, Vimentin, N1ICD, and cyclinA2 in T47D, SEN-T47D and Co-culture-T47D (which contains T47D and SEN-T47D). (**H**,**I**) Wound-healing assay of T47D subjected to the indicated treatments. (**H**) Representative micrographs of the wound-healing assay are shown. (**I**) Data represent the relative migration ratio of cells per field (error bars indicate mean SD, *n* = 3 experimental replicates, * *p* < 0.05, ** *p* < 0.01, *** *p* < 0.001). (**J**) Migration assays of T47D cells with the indicated treatments. Representative micrographs of migrated cells are shown. Data represent the number of cells derived from mean cell counts of five fields (Error bars indicate mean SD, *n* = 3 experimental replicates, ** *p* < 0.01, *** *p* < 0.001). (**K**) Invasion assay of T47D cells with the indicated treatments. Representative micrographs of migrated cells are shown. Data represent the number of cells derived from mean cell counts of five fields (error bars indicate mean SD, *n* = 3 experimental replicates, *** *p* < 0.001). (**L**) Lung metastatic nodules were confirmed by hematoxylin and eosin staining. Scale bars, 250μm. (**M**) The number of visible surface metastatic lesions in lungs was counted (error bars indicate mean SD, *n* = 6 mice for each group, * *p* < 0.05, *** *p* < 0.001).

**Figure 5 ijms-22-00849-f005:**
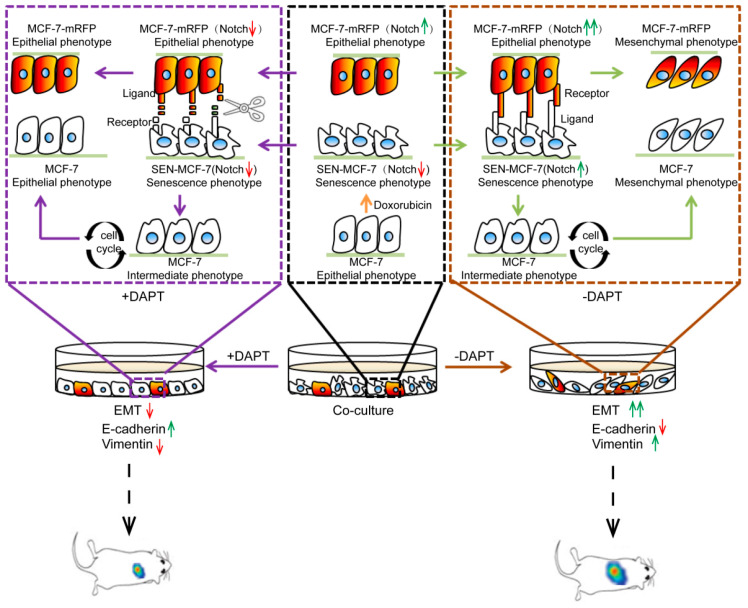
Role of Notch signaling in EMT caused by doxorubicin-induced tumor cells senescence. A proposed working model of Notch signals involved in breast cancer EMT in the co-cultivation system of senescent breast cancer cells induced by doxorubicin co-cultured with adjacent non-senescent breast cancer cells. Interactions between senescent (SEN-MCF-7) and adjacent non-senescent breast cancer cells (MCF-7-mRFP) through activating the Notch signal promote EMT of adjacent non-senescent breast cancer cells (MCF-7-mRFP) through Notch signaling, accompanied by the downregulation of E-cadherin and upregulation of Vimentin. Meanwhile, senescent breast cancer cells (SEN-MCF-7) return to the cell cycle and acquire migration and invasion ability and EMT properties, which is accompanied by the downregulation of E-cadherin and upregulation of Vimentin. Notch signaling in senescent (SEN-MCF-7) and adjacent non-senescent breast cancer cells (MCF-7-mRFP) is inhibited by treatment with DAPT in the co-cultivation system, which is accompanied by the downregulation of E-cadherin and upregulation of Vimentin in breast cancer cells. Red arrows: inhibition of signaling or downregulation of protein expression. Green arrows: activation of signaling or upregulation of protein expression.

## Data Availability

The data presented in this study are available in the Appendix A.

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
