# Peer review of "The Interaction of the Senescent and Adjacent Breast Cancer Cells Promotes the Metastasis of Heterogeneous Breast Cancer Cells through Notch Signaling"

_ijms, 2021, doi:10.3390/ijms22020849_

Round 1

Reviewer 1 Report

This study investigates the effect of tumor senescence induced by chemotherapeutic drugs on tumor progression. Using a direct co-culture model of MCF-7 breast cancer cells, the authors found a regulatory mechanism underlying the chemotherapeutic drug-mediated breast cancer recurrence that involved activation of Notch signaling. The study reports that cell migration and invasion ability of senescent breast cancer cells induced by doxorubicin treatment can be inhibited by the Notch inhibitor DAPT. In addition, in vivo studies show that inhibition of Notch signaling suppressed metastasis of senescent and adjacent non-senescent breast cancer cells. As a conclusion, the authors propose Notch inhibitors as a promising adjuvant therapy to doxorubicin for breast cancer.

The introduction is comprehensive and explains very well the state-of-the-art art. The study design is adequate and overall results are consistent and very well illustrated.

Minor point:

– The data is reinforced by the use of a second breast cancer cell line T47D that is only shown in supplementary data (in fig S3). I believe that these results are very important to support the finding of a new regulatory mechanism and should be inserted in the main manuscript.

Author Response

We greatly appreciate the reviewer’s suggestion. As shown in figure 4G-K, the data related to breast cancer cell line T47D are presented in the main manuscript.

Reviewer 2 Report

The authors present the activation of Notch signaling as a mechanism underlying breast cancer recurrence mediated by chemotherapy drugs. These findings contribute to the utility of inhibitors of the Notch pathway as adjuvant agents in combination with doxorubicin for breast cancer. The manuscript is well-written.

Results: 2.1. Senescence of breast cancer cells induced by doxorubicin promotes EMT of adjacent breast cancer cells

2.2. Senescence of breast cancer cells induced by doxorubicin promotes EMT of adjacent breast cancer cells

The subjects are same. One of them should be corrected.

Author Response

We thank reviewers for pointing this out and we are sorry for the error.

“2.2. Senescence of breast cancer cells induced by doxorubicin promotes EMT of adjacent breast cancer cells” should be “Senescent breast cancer cells re-enter the cell cycle and undergo EMT after direct co-culture with breast cancer cells”. We have corrected them by modifying the text.